# Integrated Damage Location Diagnosis of Frame Structure Based on Convolutional Neural Network with Inception Module

**DOI:** 10.3390/s23010418

**Published:** 2022-12-30

**Authors:** Jianhua Ren, Chaozhi Cai, Yaolei Chi, Yingfang Xue

**Affiliations:** School of Mechanical and Equipment Engineering, Hebei University of Engineering, Handan 056038, China

**Keywords:** frame structure, fault diagnosis, convolutional neural network, anti-noise ability, inception

## Abstract

Accurate damage location diagnosis of frame structures is of great significance to the judgment of damage degree and subsequent maintenance of frame structures. However, the similarity characteristics of vibration data at different damage locations and noise interference bring great challenges. In order to overcome the above problems and realize accurate damage location diagnosis of the frame structure, the existing convolutional neural network with training interference (TICNN) is improved in this paper, and a high-precision neural network model named convolutional neural network based on Inception (BICNN) for fault diagnosis with strong anti-noise ability is proposed by adding the Inception module to TICNN. In order to effectively avoid the overall misjudgment problem caused by using single sensor data for damage location diagnosis, an integrated damage location diagnosis method is proposed. Taking the four-story steel frame model of the University of British Columbia as the research object, the method proposed in this paper is tested and compared with other methods. The experimental results show that the diagnosis accuracy of the proposed method is 97.38%, which is higher than other methods; at the same time, it has greater advantages in noise resistance. Therefore, the method proposed in this paper not only has high accuracy, but also has strong anti-noise ability, which can solve the problem of accurate damage location diagnosis of complex frame structures under a strong noise environment.

## 1. Introduction

With the continuous development of productivity in China, frame structure is widely used in a series of fields such as machinery, civil engineering, aerospace and medicine [1,2,3,4]. The health of a frame structure directly affects the normal operation of machinery, the safety and stability of civil structure, the stability of the model state and the safety of people’s lives. Therefore, it is of great practical significance to carry out health monitoring and fault diagnosis on the frame structure, find hidden dangers in the operation process of the frame structure in a timely manner and evaluate its safety and subsequent maintenance. The health monitoring and fault diagnosis of frame structures are often carried out based on vibration data. The common method is to install the acceleration sensor at the designated position of the frame structure, then collect the vibration data and use appropriate methods to achieve the health monitoring and fault diagnosis, so as to ensure the normal operation of machinery, the stability of buildings, aviation safety and emergency treatment [5].

Structural health monitoring and fault diagnosis is a hot research field that is of concern to the academic community. Scholars have proposed many effective methods in long-term scientific research and engineering practice, and have made fruitful achievements in applying these methods to structural health monitoring and fault diagnosis. Early structural health monitoring and fault diagnosis tasks are relatively simple, so Fourier transform [6,7] and wavelet analysis [8,9] are applied to research and obtain good results. With the complexity of structural health monitoring and fault diagnosis tasks and the development of technology, scholars have applied principal component analysis (PCA) [10,11], support vector machine (SVM) [12,13], artificial neural network (ANN) [14,15] and other methods, and have also made many research achievements. With the continuous progress of science and technology, especially the arrival of the big data era, new problems and challenges are brought to the field of structural health monitoring and fault diagnosis, which makes it more and more difficult to use the above traditional methods for structural health monitoring and fault diagnosis. 

With the rapid development of computer technology, the processing capacity of computers has been greatly improved, which has led to the rapid development of artificial intelligence. Thus, deep learning theory that can extract features from massive data has emerged. With its powerful data mining and autonomous learning ability, deep learning has made major breakthroughs in image processing [16,17,18], speech recognition [19,20,21], machine vision [22,23,24], text processing [25,26,27] and other fields. With the continuous development of deep learning theory, it has also been widely studied and applied in mechanical equipment fault diagnosis and structural health monitoring. In the research of mechanical equipment fault diagnosis, the fault diagnosis of rotating machinery has always been one of the hot spots. Therefore, fault diagnosis of rotating machinery based on deep learning theory has been deeply studied. In order to improve the accuracy and anti-noise ability of bearing fault diagnosis, Zhang et al. proposed a network model named convolutional neural network with training interference (TICNN) based on a convolutional neural network [28]. In order to embed domain diagnosis knowledge into deep learning to obtain appropriate features related to good health, Chen et al. proposed a rolling bearing fault diagnosis method based on cyclic spectral coherence and convolutional neural network [29]. Wang et al. proposed a general bearing fault diagnosis model based on the Alexnet model, which can reduce prior knowledge and extra time [30]. In order to make up for the defect of a small number of samples, Saufi et al. designed a depth learning model based on Stacked Sparse Automatic Encoder model to process limited sample data, and developed a gearbox fault diagnosis system based on time-frequency image pattern recognition [31]. In order to avoid manual feature extraction and improve the generalization ability of the model, Singh et al. proposed a new domain adaptive method based on depth learning for fault diagnosis of a gearbox under significant speed changes [32].

Deep learning theory has also been extensively studied and applied in the aspect of structural health monitoring. In order to solve the inaccuracy caused by data anomaly in structural health monitoring, Bao et al. proposed a data anomaly detection method based on computer vision and deep learning [33]. In order to overcome the shortcomings of image-based structural prognostics and health monitoring algorithms, Sarkar et al. proposed a damage detection method based on depth learning technology to characterize the crack form on composite materials [34]. In order to realize damage identification and location in real large-scale systems, Azimi et al. proposed a new structural health monitoring method based on transfer learning technology and convolutional neural network [35]. To solve the problem of data anomaly in the data preprocessing stage of structural health monitoring, Tang et al. proposed a data anomaly detection method based on convolutional neural network [36]. Abdeljaber et al. conducted fault diagnosis research on a grandstand frame and four-story frame structure by using a one-dimensional revolution neural network (1DCNN) in 2017 and 2018, respectively [37,38]. However, it has poor anti-noise ability and needs a lot of manual data processing. Therefore, it is necessary to propose better methods to reduce manual data processing and improve anti-noise ability, so as to achieve high-precision fault diagnosis of frame structures.

The inverse problem of structural damage identification is also a research hotspot in the field of structural health monitoring. The inverse problem of structural damage identification based on deep learning has also been deeply studied and widely applied. In order to determine and identify the damage load parameters of structures and materials, Ren et al. developed a new depth learning computing framework based on the permanent or residual plastic deformation distribution or damage state of structures; they proposed an artificial intelligence (AI) inverse problem solution for failure analysis [39]. Rautela et al. used convolutional neural network and recurrent neural networks to solve the inverse problem of numerical approximation, and proposed a hybrid strategy of classification and regression in a supervised environment for combined damage detection and location [40]. In order to solve the multiparameter underdetermined inverse problems that are at the core of nonlinear ultrasonic technique and create a high fidelity physics-based model, Liu et al. introduced the size and location information of the damage into the assembled dataset, and proposed a nonlinearity-aware discrete wavelet transform-bidirectional long short-term memory network [41]. Chen et al. developed a data-driven AI inverse problem solution for traffic collision reconstruction based on a deep neural network to determine and identify the initial collision conditions of vehicle collision [42].

A frame structure needs to be repaired in time when the damage degree reaches a certain dangerous value, so the damage location of the frame structure needs to be directly diagnosed for maintenance. In addition, engineers also need to analyze the impact of damage at different locations of the frame structure on the overall stability of the frame structure, and determine the components that can easily damage the balance of the frame structure, so as to carry out purposeful, targeted and accurate maintenance and repair of the frame structure. The damage probability obtained through the diagnosis of the damage degree of the frame structure can indirectly obtain the damage location of the frame structure; however, when the noise interference is strong, the adjacent damage degree values will become closer due to uncertainty factors, so it is difficult to make a very accurate damage location identification of the frame structure. In order to solve the above problems and overcome the shortcomings of traditional fault diagnosis methods, and improve the accuracy of damage location diagnosis of frame structure under a strong noise environment, the existing TICNN is improved in this paper, and a high-precision fault diagnosis neural network model with strong anti-noise ability is proposed by adding the Inception module to the network [43]. In addition, in order to effectively avoid the overall misjudgment problem caused by using single sensor data for damage location diagnosis, an integrated damage location diagnosis method is proposed. Taking the four-story steel frame model [44] of the University of British Columbia as the research object, the method proposed in this paper is tested and compared with other methods. The experimental results show that the accuracy of the proposed method is 97.38%, which is higher than other methods; at the same time, it has greater advantages in noise resistance.

The following specific contents of this paper are as follows. Section 2 introduces the structural composition and damage cases of the test object. The idea and specific process of integrated damage location diagnosis and the preprocessing of training data are introduced in Section 3. Section 4 presents the improved TICNN named convolutional neural network based on Inception (BICNN) and analyzes its performance. The integrated damage location diagnosis results of the frame structure are obtained in Section 5. Section 6 summarizes this paper and draws relevant conclusions. 

## 2. Description of Research Object

The research object and data of this study are from the four-story steel frame structure of the Seismic Laboratory of the University of British Columbia. The frame structure is shown in Figure 1 [44], with a height of 3.6m and a square bottom of 2.5 m × 2.5 m. It is composed of four faces: east, south, west and north. Since the structural distribution of each location on the four-story steel frame structure is the same, and the materials and dimensions are the same, the structures at the same location on the four surfaces were marked with the same number (1~12). In order to obtain the vibration data representing the damage of the frame structure, three acceleration sensors were installed on each floor starting from the first floor: one acceleration sensor was in the west, one acceleration sensor was in the east, and the other acceleration sensor was near the central column. Therefore, a total of 15 sensors were used to collect vibration data, which were numbered 1~15. The acceleration sensors numbered 1~3 were placed at the ground base of the bottom layer, and the rest were placed on the top of each floor. When the structures at some locations on the frame structure were damaged, the vibration signal collected by the above acceleration sensor could be used to diagnose the damage location.

In order to simulate the damage of the frame structure and realize the diagnosis of its damage degree and damage location, nine types of damage location were obtained by removing and loosening the structures numbered 1~12 on the four surfaces in Figure 1. Nine damage cases are shown in Table 1.

## 3. Integrated Damage Location Diagnosis Processes

Because there are always some stable locations in the frame structure under different damage cases, there is a large error in selecting the data collected by the acceleration sensor at the stable location for damage location diagnosis. In addition, the number of beams and columns constituting the frame structure is large, leading to the installation of more sensors and damage cases, which increases the difficulty of accurate damage location diagnosis of the frame structure. In order to solve the above problems and effectively avoid the overall misjudgment problem caused by using single sensor data for damage location diagnosis, this paper proposes an integrated damage location diagnosis method based on BICCN. In this method, firstly, the network model is trained with the selected data of multiple sensors at the same time to form multiple classification models; then, the trained models are used to classify the data of multiple acceleration sensors at the same time, and the category of the maximum average value of multiple classification results is taken as the damage location category. The integrated damage location diagnosis process of the frame structure is shown in Figure 2.

In Figure 2, $C_{i}$ is the *i*-th damage case; $DC_{ij}$ is the number of signals of the *j*-th acceleration sensor in the *i*-th case after data enhancement; $DN_{ijt}$ is the normalization result of the *t*-th segment of the *j*-th acceleration sensor under case *i*; ${\mathrm{ACC}}_{j}$ is the accuracy obtained by training with the *j*-th acceleration sensor data; ${\mathrm{LOSS}}_{j}$ is the loss obtained by training with the *j*-th acceleration sensor data; $P_{ij}$ is the probability that 20% of the signals of the *j*-th acceleration sensor not participating in the training are judged to be the *i-*th damage case; $\bar{P_{i}}$ is the average value of $P_{ij}$ under damage case *i*.

(1)Data collection

In the frame structure model of the experimental object, a total of nine kinds of damage cases were carried out. Each damage case can be represented by $C_{i}$, where *i* = 1~9. The vibration signals collected by 12 vibration sensors installed on the frame structure can be expressed as follows:(1)$$C_{i}=[C_{i4}{,C}_{i5},\cdot \cdot \cdot C_{ij},\cdot \cdot \cdot C_{i15}]$$
where *j* is the *j*-th acceleration sensor. Starting from the bottom layer of the frame structure, three acceleration sensors were placed on each layer to collect vibration signals. Because the data collected by the acceleration sensors on the bottom layer are less different when the frame structure is damaged, taking 12 acceleration sensors numbered 4–15 as the data source can effectively reduce the damage location diagnosis error, so *j* = 4~15.

(2)Low pass filtering

In most environments, the noise that interferes with the frame structure is Gaussian noise [45], so Gaussian low-pass filter was used to filter the original vibration signal. The Gaussian distribution is as follows:(2)$$G(x)=\frac{1}{\sqrt{2\pi }\sigma }e^{-\frac{x^{2}}{2\sigma^{2}}}$$
where x is the signal to be filtered; σ is the variance of the signal.

The Gaussian filter can select the weight according to the shape of the above Gaussian function so as to smooth the data image. Since the mean value of Gaussian white noise is 0, the mean value in the Gaussian filter was set to 0, so the mean value was not shown in the equation. At the same time, in order to filter enough white noise in the vibration signal, the variance in the Gaussian filter was set as the variance of the vibration signal.

(3)Data enhancement and clipping processing

In order to increase the number of training samples, this paper used the sliding window method to enhance and clip the original data. The signal segments with different length can be obtained by adjusting the size of the window, and the signal segments with different number can be obtained by adjusting the sliding step size of the window. The sliding window method can obtain enough training samples to the greatest extent, and the training data obtained by aperiodic sampling can improve the generalization ability of the network. The calculation formula of data enhancement is as follows:(3)$$DC_{ij}=\frac{C_{ij}-l}{b}$$
where $C_{ij}$ is the number of signals of the *j*-th acceleration sensor in the *i*-th case, l is the size of the sliding window and $DC_{ij}$ is the number of signals of the *j*-th acceleration sensor in the *i*-th case after data enhancement. It can be seen that the maximum number of samples that can be obtained is $C_{ij}-l$, when the sliding step *b* is equal to 1. According to the data provided by the University of British Columbia, the length of vibration signals in cases 1~5 is 24,000, the length of vibration signal in case 6 is 60,000, and the length of vibration signals in cases 7~9 is 72,000. Therefore, in this paper, the window size is 1024, the step *b* is 24 in cases 1~5, the step *b* is 60 in case 6, and the step is 72 in cases 7~9. Therefore, it can be seen that the number of data segments clipped from the data collected by each acceleration sensor in each case is 958.

(4)Normalization processing

Normalizing the data between [−1, 1] can ensure that the data information is not lost. The normalization formula is as follows:(4)$$DN_{ijt\;(t\;=\;1,\;2,\;...,\;n)}=\frac{DC_{ijt}-average(DC_{ij})}{\left|\max(DC_{ij})\right|}$$
where $DN_{ijt}$ is the normalization result of the *t*-th segment of the *j*-th acceleration sensor under case *i*.

(5)Data set partition

After clipping and normalization, 80% data of each acceleration sensor in nine cases were labeled and randomly mixed into a data set. In 80% of the data, 70% of the samples were used as training set, and the other 30% were test set. The remaining 20% of the data were used for verification experiment of damage location diagnosis. The sample size corresponding to each convolutional neural network is shown in Table 2.

(6)Selection of training data

Because it is cumbersome to train 12 groups of data and use 12 groups of $DN_{ij}$ for damage location diagnosis of a frame structure, and correct damage location diagnosis can be made by using one group or several groups at the same time, selecting several groups from 12 groups of $DN_{ij}$ with fast convergence speed and high recognition rate for integrated damage location diagnosis can not only improve the diagnosis efficiency, but also ensure the reliability of the diagnosis results. Here, it is assumed that *n* groups with excellent vibration signals were selected from 12 groups’ vibration signals.

(7)Calculation of damage location probability

The trained convolutional neural network model is used to predict the $DN_{ij}$ of the vibration signals obtained by *n* sensors that are not involved in the training task of neural network, so as to calculate the damage location probability. The calculation formula is as follows:(5)$$P_{ij}=\frac{p_{i}}{DN_{ij}\times 20\%}$$
where $P_{ij}$ is the probability that 20% of the signals of the *j*-th acceleration sensor not participating in the training are judged to be the *i-*th damage case, and $p_{i}$ is the number of the *i*-th damage case judged by the classifier in 20% of $DN_{ij}$.

(8)Comprehensive damage location probability

In order to avoid misjudgment caused by excessive interference, the maximum value of $P_{ij}$ cannot be used as the classification standard of damage cases, but the maximum average value of $P_{ij}$ should be used to specify the damage cases. The average value of $P_{ij}$ can be calculated as follows:(6)$$\bar{P_{i}}=\frac{1}{n}\sum_{j=1}^{n}P_{ij}$$

## 4. BICNN 

This paper improved the topology of TICNN and proposed a new network model named BICNN. Because the recognition rate was low when using TICNN to diagnose the damage location of the frame structure, the depth of TICNN was increased, which increased the depth of feature extraction of the neural network and improved the ability of feature extraction, thus improving the recognition rate of TICNN. On this basis, the small convolution layer of TICNN was replaced with the Inception module, which increased the scale of convolution. Because the feature information obtained by convolution kernels of different sizes is different, the feature parameters extracted by feature superposition are more comprehensive, which further improved the recognition rate of TICNN. Therefore, a new neural network based on TICNN and Inception module named BICNN was proposed. The basic principle of BICCN is: first, after the first convolution layer, Dropout [46] is used to make the convolved output data randomly inactivate according to probability *P*, so as to improve the network’s ability to resist data loss; at the same time, Dropout is also used at the full connection layer to enable the network to avoid over fitting; then, after two convolution layers, five groups of Inception modules are used to improve the information recognition ability; finally, BN [47] (batch normalization) is used after each convolution layer to improve the anti-noise ability of the network. The structure of BICNN is shown in Figure 3.

In Figure 3, because the parameters and structure of each inception module are the same, the other four Inception modules are simply described with “Inception module × 4”. The classifier of BICNN is Softmax, and the optimizer is Adam.

### 4.1. Inception Module

Inception module is a module in Google-net. Its basic principle is to input the output of the previous layer to four convolution layers of different sizes for convolution operation, and then add them after BN processing, so as to enhance the fitting ability of the network. The use of the inception module can improve the utilization of parameters, and the use of convolution cores of different sizes can increase the diversity of recognition contents. At the same time, the use of convolution cores with size = 1 can save parameters, accelerate operation, suppress over fitting and increase the ability of nonlinear expression. The schematic diagram of inception module is shown in Figure 4 overfit.

Since the output of each Inception module is connected to a maximum pooling layer with size = 2 and step = 2, the input in the above figure is the output of the previous Inception module after the maximum pooling. The schematic diagram of using BICCN to diagnose the damage location of the frame structure is shown in Figure 5.

### 4.2. Parameters Design

Because the BICNN proposed in this paper is obtained by improving TICNN, and reference [28] has proven that TICNN has a higher recognition rate and faster training speed, the main parameters of BICNN are all from TICNN. Since the input data length of TICNN is 2048 and the input data length of BICNN is 1024, in order to make the feature image size after convolution of the first convolution layer consistent, the convolution step of the first convolution layer was reduced from 16 to 8, so as to continue the high recognition rate and high anti-interference performance of small convolution layers in TICNN. In order to increase the separation ability of the network, this paper adopted three full connection layers to match classifier to increase classification accuracy. In addition, you only need to set the number of channels in Inception, and its convolution kernel size is shown in Figure 4. Specific parameters of BICNN can be seen in Table 3.

### 4.3. Selection of Training Samples

It can be seen from Section 3 that the integrated damage location diagnosis of a frame structure proposed in this paper involves selecting several groups of excellent sensor data from the data collected by 12 vibration sensors for diagnosis research. Therefore, selecting excellent sensor data as network training samples is the premise of realizing damage location accuracy diagnosis of frame structures. In order to select excellent network training samples, the BICCN model proposed in this paper was used to conduct experimental research on data from 12 acceleration sensors under nine damage cases, and samples suitable for damage location diagnosis of the frame structure were obtained. In order to ensure the reliability of the selected samples, 1DCNN, first-layer wide convolution kernel deep convolutional neural network (WDCNN), TICNN and improved convolutional neural network with training interference (ITICNN) [48] were used for experimental research.

In this paper, the specific configuration of the laptop for the experimental study is as follows. CPU: Intel Core i7-4710MQ, GPU: NVIDIA GT940M 2G and memory: 12G. During training, the data processing and convolutional neural network operating environments were Tensorflow 2.4.8, keras 3.4.2 and Python 3.8.0; the initial learning rate was the default value of Adam in the operating environment; the batch size was 64. Since the main purpose of this section is to select training samples that make the network converge quickly and have high accuracy, the training epoch in this section is 150, and the training epoch in the subsequent noise resistance comparison experiment and actual diagnosis is 600. The recognition accuracy of the above five networks is shown in Table 4.

It can be seen from Table 4 above that the excellent training samples separated by 1DCNN is 0, while WDCNN, TICNN, ITICNN and BICNN have separated five groups’ training samples with high recognition accuracy; these training samples are from acceleration sensors numbered 4, 5, 6, 9 and 14. Therefore, the vibration signals collected by acceleration sensors numbered 4, 5, 6, 9 and 14 were used as training samples for integrated damage location diagnosis of frame structures in this paper, so as to achieve accurate damage location diagnosis of frame structures. In addition, it can be seen from the data in Table 4 that the BICCN network model proposed in this paper is superior to 1DCNN, WDCNN, TICNN and ITICNN in accuracy.

### 4.4. Comparison of Anti-Noise Ability

In order to verify the superiority of the BICCN network model proposed in this paper in the aspect of anti-noise, comparative experiments were conducted with 1DCNN, WDCNN, TICNN and ITICNN under the same experimental conditions. In the experiment, firstly, Gaussian white noise of different intensity was added to the test set, and then the anti-noise ability of different network models was compared according to the training accuracy of the test set after training.

The generation of Gaussian white noise is as follows.

(1) Use the Randn function provided by Tensorflow to generate a set of series marked *Y* whose length is consistent with the length of training sample, and whose value is between [−1, 1] and follows the standard normal distribution
(7)$$Y=[Y_{1},Y_{2},\cdot \cdot \cdot \cdot \cdot \cdot \cdot \cdot \cdot \cdot Y_{n}]$$

(2) Add noise signals of different intensity to the test set
(8)$${\overset{\wedge }{X}}_{i}=X_{i}+kY_{i}$$
where $X_{i}$ is the signal of the test set; *k* is the adjustment coefficient of noise intensity.

Then the intensity of the noise signal added to the test set is
(9)$$P_{noise}=\sum_{i=1}^{n}{\left(kY_{i}\right)}^{2}=k^{2}\sum_{i=1}^{n}{\left(Y_{i}\right)}^{2}$$
where $k^{2}\sum_{i=1}^{n}{\left(Y_{i}\right)}^{2}$ the white noise obeying the standard normal distribution and the power is equal to 1. Therefore, the noise intensity added to the test set is $k^{2}$, and changing the value of *k* can change the intensity of white noise added to the test set, so as to achieve the purpose of adding different noises to the test set.

It can be seen from Equation (7) to Equation (9) that the noise sequence generated is a standard normal distribution between [−1, 1], and the training samples are normalized to [−1, 1]. Therefore, when *k* = 1, the amplitude of the noise added to the training samples is equal to the amplitude of the training samples; when *k* = 0.5, the amplitude of the noise added to the training sample is 0.5 times the amplitude of the training sample, and so on.

To verify the superiority of the BICNN model proposed in this paper in terms of anti-noise, firstly, different intensity of white noise was added to the data of nine damage test sets collected by the acceleration sensor numbered 9 by changing the value of *k*, and then used the models trained and saved by 1DCNN, WDCNN, TICNN, ITICNN and BICNN to classify the signals with different noises. The accuracy of the five neural network models is shown in Table 5.

As can be seen from Table 5, with the continuous increase of *k*, the accuracy of each neural network model is declining, but the decline speed of the accuracy of BICNN is slower than that of other models. When *k* is equal to 0.3, the accuracy of ITICNN and BICNN is more than 90%, which is much higher than the accuracy of the other three network models. When *k* reaches 0.5, the accuracy of BICNN can still reach 77.03%, and when *k* increases to 1, the accuracy of BICNN is 36.17%, which is higher than the accuracy of ITICNN, TICNN, WDCNN and 1DCNN. Therefore, the BICNN model proposed in this paper has a strong anti-noise ability, and its anti-noise performance is better than ITICNN, TICNN, WDCNN and 1DCNN.

### 4.5. Comparison of Fitting Ability

In addition to comparing the anti-noise performance of different neural network models with fault diagnosis accuracy, the fitting ability of the neural network model can be measured by observing the change of Loss (objective function value) during the training of convolutional neural network. In the training process, the lower the Loss is, the stronger the fitting ability of the network is. The Loss curves of 1DCNN, WDCNN, TICNN, ITICNN and BICNN are shown in Figure 6.

In Figure 6, BICNN-T-Loss is the change curve of Loss obtained by training with BICNN; BICNN-V-Loss is the change curve obtained by testing with BICNN. When 1DCNN, TICNN, WDCNN and ITICNN were used for training and testing, the change curve of Loss obtained also adopted the same expression. As shown in Figure 6, five networks are used to train the data collected by the acceleration sensor numbered 9 for 600 epochs at the same time, and 400~600 epochs are taken for comparison. When using 1DCNN for training and testing, Loss experienced three large fluctuations with amplitude greater than 0.09. When using WDCNN for training and testing, Loss experienced one large fluctuation with amplitude greater than 0.018. When using TICNN for training and testing, Loss experienced five large fluctuations; the maximum amplitude of Loss is about 0.033, and the minimum amplitude is about 0.015. When using ITICNN for training and testing, Loss experienced one large fluctuation with amplitude greater than 0.08. When using BICNN for training and testing, Loss experienced three large fluctuations with amplitude of 0.011, 0.018 and 0.011. Therefore, from the above data, it can be seen that compared with 1DCNN, TICNN and ITICNN, the Loss curves of WDCNN and BICNN are relatively stable and have stronger fitting ability.

### 4.6. Visualization Analysis of BICNN’s Performance

TSNE cluster is a commonly used method to reduce dimensions. It can convert the output dimensions of specified layers, such as full connection layer, into a one-dimensional scatterplot, and the changes of data sets in the network can be intuitively shown. In order to more intuitively express the damage location diagnosis results of structural frame, TSNE cluster was used to visualize the diagnosis results. Under the condition of no noise, with the data collected by the acceleration sensor numbered 9 as the research samples, the BICNN model proposed in this paper was used to classify the nine damage cases, and TSNE cluster was used to visualize the output of the full connection of the last layer of BICNN; the visualization results are shown in Figure 7.

As can be seen from Figure 7, the visualization diagnosis results are shown when the training epoch is equal to 3, 150, 300 and 500, respectively. It can be seen from the visualization results that when the training epoch is equal to 3, the nine damage cases are completely mixed together, and the network model cannot distinguish them at all; when the training epoch is equal to 150, part of the nine damage cases can be effectively identified by the network model, which shows that the network model has certain classification ability at this time; when the training epoch is equal to 300, nine damage cases can be accurately identified by the network model; when the training epoch is equal to 500, each damage case shrinks spontaneously, which increases the distance between different cases, indicating that the network model at this time has stronger recognition ability and can meet the requirements of accurate damage location diagnosis of frame structures.

## 5. Integrated Diagnosis Results of Damage Location

In this section, five BICNN models were used for integrated damage location diagnosis of a frame structure. In order to verify the superiority of BICNN in anti-noise, 1DCNN, WDCNN, TICNN and ITICNN were used for comparative experimental research under different noise conditions.

### 5.1. Diagnosis Results of BICNN without Noise

In order to make the samples of integrated damage location diagnosis and the samples of training set and test set not repeat, 96 signal segments with length of 1024 were cut out according to step = 240 for damage cases 1~5, step = 600 for case 6 and step = 720 for cases 7~9, and the five BICNN models (the original number of BICNN corresponds to the selected sensor, namely 4, 5, 6, 9 and 14) were renumbered as 1~5 to facilitate the expression of diagnosis results. *P_ij_* can be calculated by inputting these signal fragments into the corresponding BICNN model for prediction. The probabilities (*P_ij_*) predicted by five BICNN models for nine damage cases are shown in Figure 8.

It can be seen from Figure 8 that when the data collected by the selected five acceleration sensors are used to classify the damage locations, the data collected by the acceleration sensors numbered 4 can only correctly classify the damage cases 1~4, but misjudgment occurs when classifying the damage cases 5~9. The probability of misjudgment of the data collected by the acceleration sensors numbered 5, 6, 9 and 14 during training and testing is very low. The maximum value of misjudgment occurs in the diagnosis of damage case 1 using the data collected by the acceleration sensor numbered 5. The misjudgment probability is 20%, of which 19% is judged as damage case 5 and 2% as damage case 4. After calculation, the average value $\bar{P_{i}}$ of *P_ij_* can be obtained, as shown in Figure 9.

It can be seen from Figure 9 that in damage cases 5~9, $\bar{P_{i}}$ of damage case 4 are relatively large, which indicates that misjudgment occurred in the classification of nine damage cases. It can be seen from Figure 8 that the misjudgment is caused by the insufficient classification accuracy when the data collected by the acceleration sensor numbered 4 is used for classification. The integrated damage location diagnosis accuracy can be calculated according to the data in Figure 8. The diagnosis accuracy formula is as follows:(10)$$A=\frac{\sum_{i=1}^{9}{\left(Md\right)}_{i}}{9}\times 100\%$$
where ${\left(Md\right)}_{i}$ represents the value of the *i*-th element on the main diagonal in Figure 9. Through calculation, the integrated diagnosis accuracy of BICNN is 86.42%. Because the classification accuracy of the data collected by the acceleration sensor numbered 4 is not high, the data collected by the acceleration sensor 4 was deleted when the damage location of the frame structure was diagnosed, and the accuracy of the integrated diagnosis of the damage location of the structure frame by using BICNN is 97.38% after recalculation, which achieves satisfactory results.

### 5.2. Influence of Noise on Diagnosis Results

In Section 4.4, the anti-noise ability of five convolutional neural networks in the training process was studied. This section focuses on the anti-noise ability of five convolutional neural networks in integrated damage location diagnosis. During the experiment, firstly, noise signals with different intensities were added to the vibration signals collected by four acceleration sensors (the data collected by the acceleration sensor numbered 4 were removed) under nine kinds of damage cases. Then, five convolutional neural networks (1DCNN, WDCNN, TICNN, ITICNN and BICNN) were used to conduct integrated diagnosis on the damage location of the frame structure. Their anti-noise ability was compared according to the integrated diagnosis accuracy of five convolutional neural networks. The comparison results are shown in Table 6.

It can be seen from the above table that when the noise added to the signal to be measured is higher than 50%, TICNN cannot meet the minimum requirements for effective recognition; WDCNN cannot meet the minimum requirements for effective recognition when *k* = 0.8; 1DCNN cannot extract effective information when *k* = 0.6; and ITICNN and BICNN still have a certain diagnosis accuracy until the intensity of noise signal reaches 100% (*k* = 1) and the integrated diagnosis accuracy are higher than 11.11%. Compared with BICNN and ITICNN, it can be seen that the integrated diagnosis accuracy of BICNN is higher than that of ITICNN, which shows that BICNN has higher reliability in the case of large noise. In addition, it can be seen from Table 6 that when there is no noise, the BICNN proposed in this paper has the highest integrated diagnosis accuracy and is superior to other convolutional neural network models in comparison. Therefore, the BICNN proposed in this paper not only has higher integrated accuracy than other similar convolutional neural models, but also has obvious advantages in anti-noise. It can be used as a convolutional neural network model for accurate damage location and diagnosis of frame structures under a strong noise environment.

### 5.3. Discussion on Diagnosis Results

In order to solve the problem of accurate damage location diagnosis of frame structures, a new convolutional neural network model named BICNN was proposed based on the improvement of TICNN. In order to verify the performance and advantages of the model proposed in this paper, a comparative experimental study was carried out with 1DCNN, WDCNN, TICNN and ITICNN under the same experimental conditions. It can be seen from the comparative results of the sensor optimization experiment shown in Table 4 that when the data of a single sensor is used for classification, the BICNN proposed in this paper can achieve accurate classification for each sensor data, and the classification accuracy is higher than that of 1DCNN, WDCNN, TICNN and ITICNN. This shows that the model proposed in this paper has higher classification accuracy and is superior to other four methods mentioned above. It can be seen from the anti-noise comparative experiment results shown in Table 5 that when there is no noise and the noise intensity is relatively low, the above five models can achieve accurate classification, and the accuracy of all models reached 100%. With the increase in noise intensity, the accuracy of each model decreased, but the accuracy of the model proposed in this paper decreased most slowly, and the classification accuracy under each noise level is higher than that of 1DCNN, WDCNN, TICNN and ITICNN. This shows that the model proposed in this paper is superior to the above four methods in terms of anti-noise. It can be seen from the Loss curve results of the above five models, as shown in Figure 6, that when using the model proposed in this paper and WDCNN, the amplitude of Loss curve is small and relatively stable. This shows that compared with 1DCNN, TICNN and ITICNN, BICNN and WDCNN have strong fitting ability.

In order to effectively avoid the overall misjudgment problem caused by using single sensor data for damage location diagnosis, an integrated damage location diagnosis method was proposed. In order to verify the effectiveness and superiority of the integration method proposed in this paper, under the same experimental conditions, comparative experiments were carried out with 1DCNN, WDCNN, TICNN and ITICNN, and the anti-noise ability of the model was analyzed. It can be seen from the comparative experimental results in Table 6 that when using BICNN in a noiseless environment, the integrated diagnosis accuracy is much higher than that of 1DCNN, WDCNN, and TICNN, and also higher than that of ITICNN. With an increase in noise intensity, the accuracy of all models decreased, but the accuracy of BICNN decreased the slowest; when *k* = 1, the accuracy is still 33.55%.This shows that the model proposed in this paper is not only superior to the other four methods in accuracy, but also has the strongest anti-noise ability.

## 6. Conclusions

In order to realize the accurate diagnosis of damage location of a frame structure under a strong noise environment, a high-precision fault diagnosis neural network model with strong anti-noise ability was proposed by improving the existing TICNN and adding the Inception module to the network. In order to effectively avoid the overall misjudgment problem caused by using single sensor data for damage location diagnosis, an integrated damage location diagnosis method was proposed. In order to verify the effectiveness and superiority of the method proposed in this paper, taking the four-story steel frame model of the University of British Columbia as the research object, the method proposed in this paper was tested and compared with other methods. The following conclusions can be drawn.

(1) The diagnosis accuracy obtained by using the data collected by different sensors to diagnose the damage location of the frame structure is different. Among the data collected by all sensors, the accuracy obtained by using the data collected by sensors numbered 5, 6, 9 and 14 to diagnose the damage location of the frame structure is relatively high. Therefore, the data collected by sensors numbered 5, 6, 9 and 14 can be used for integrated damage location diagnosis of the frame structure.

(2) The BICNN model proposed in this paper has high accuracy, strong anti-noise ability and fitting ability when diagnosing the damage location of frame structures, and is superior to 1DCNN, WDCNN, TICNN and ITICNN in these performances. Therefore, the BICNN model proposed in this paper can meet the requirements of accurate damage location diagnosis of frame structures.

(3) The method of integrated damage location diagnosis of a frame structure proposed in this paper not only has high accuracy, but also has strong anti-noise ability, which can effectively avoid misjudgment caused by using single sensor data for damage location diagnosis. Therefore, the integrated fault diagnosis method proposed in this paper can be used to accurately diagnose the damage location of frame structures. In addition, the integrated fault diagnosis method proposed in this paper can also be applied to the fault diagnosis of rotating machinery, medical health detection and other fields.

## Figures and Tables

**Figure 1 sensors-23-00418-f001:**
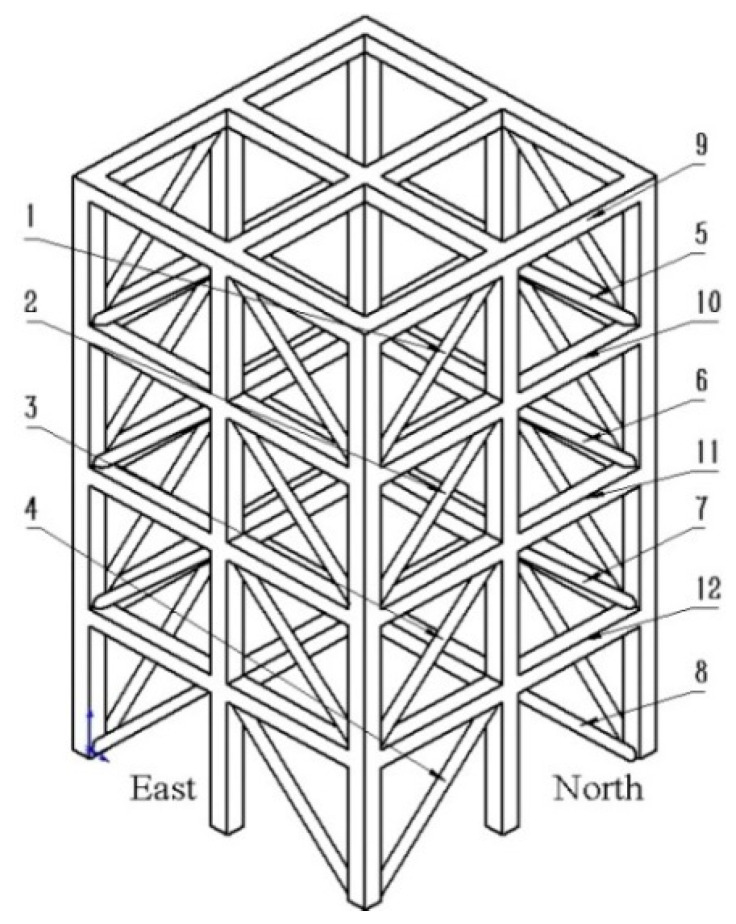
Frame structure model.

**Figure 2 sensors-23-00418-f002:**
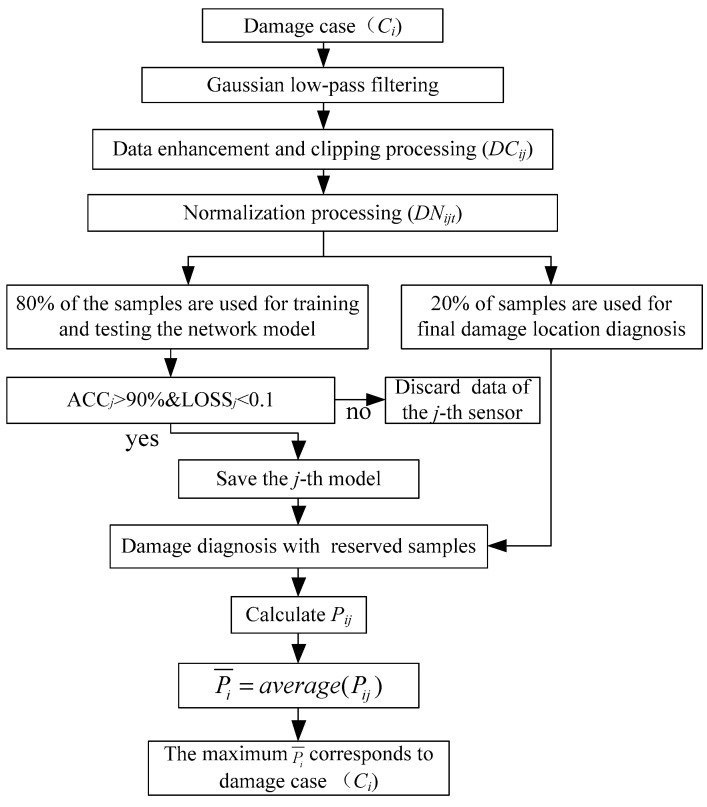
Integrated damage location diagnosis process.

**Figure 3 sensors-23-00418-f003:**
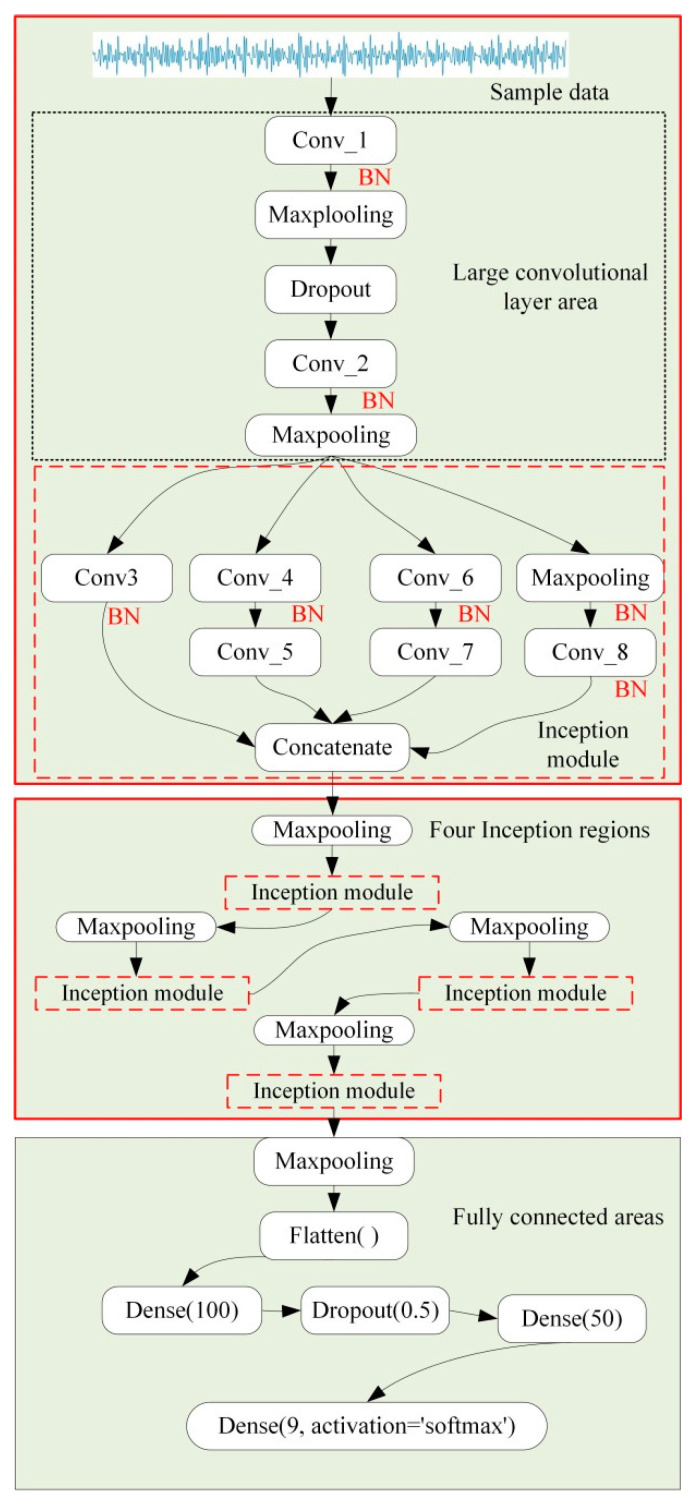
Structure of BICNN.

**Figure 4 sensors-23-00418-f004:**
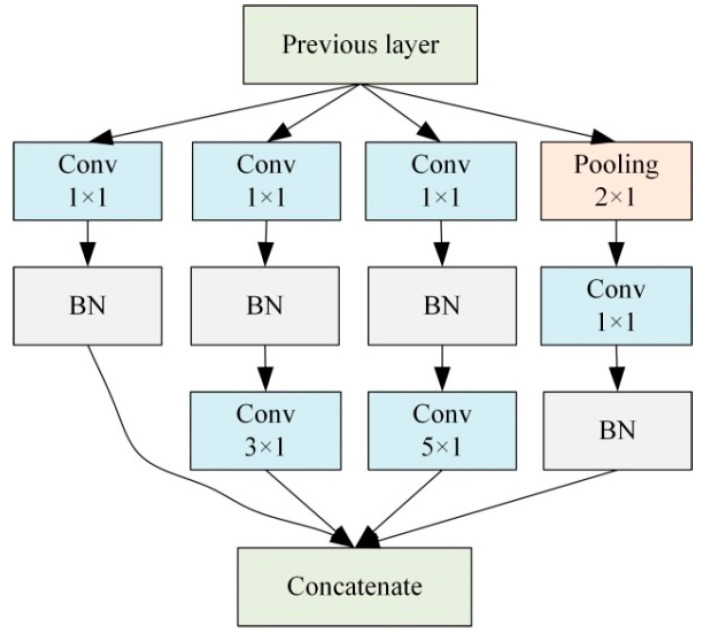
The schematic diagram of inception module.

**Figure 5 sensors-23-00418-f005:**
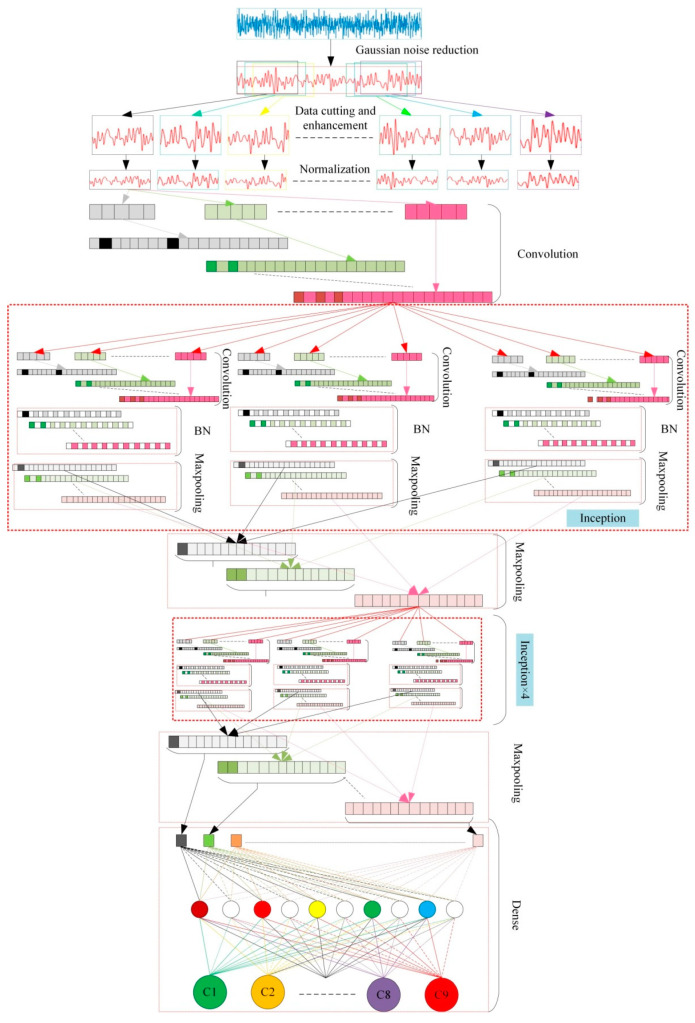
The schematic diagram of damage location diagnosis.

**Figure 6 sensors-23-00418-f006:**
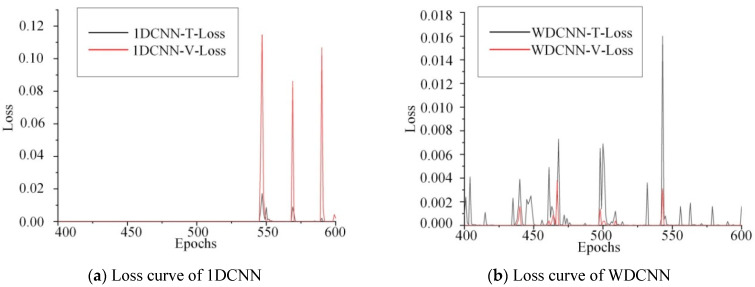
Loss curves of 1DCNN, WDCNN, TICNN, ITICNN and BICNN.

**Figure 7 sensors-23-00418-f007:**
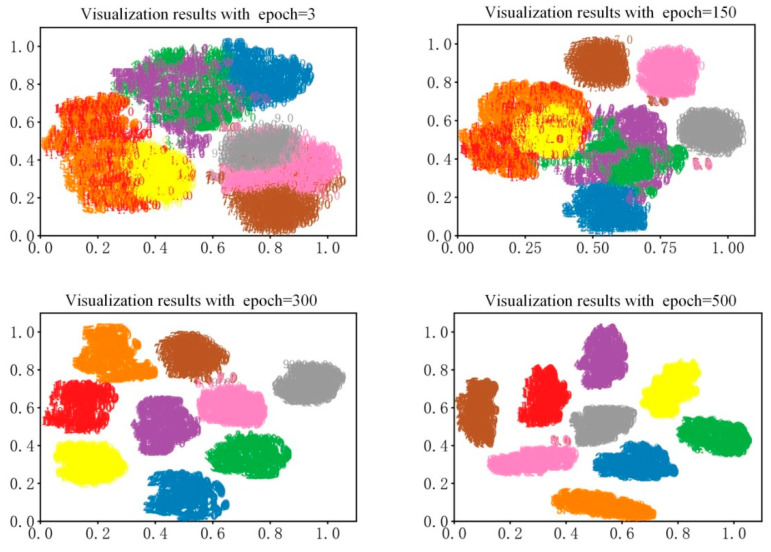
Visualization results of BICNN.

**Figure 8 sensors-23-00418-f008:**
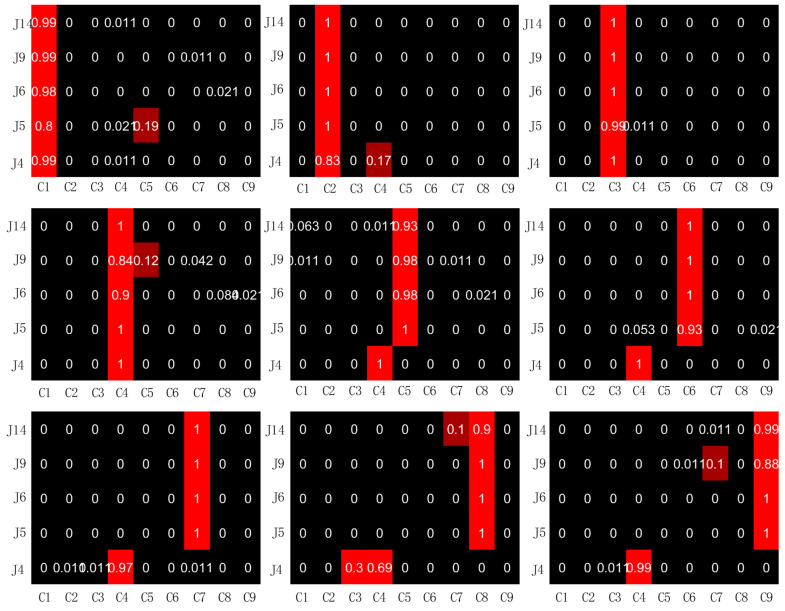
*P_ij_* predicted by 5 BICNN models for 9 damage cases.

**Figure 9 sensors-23-00418-f009:**
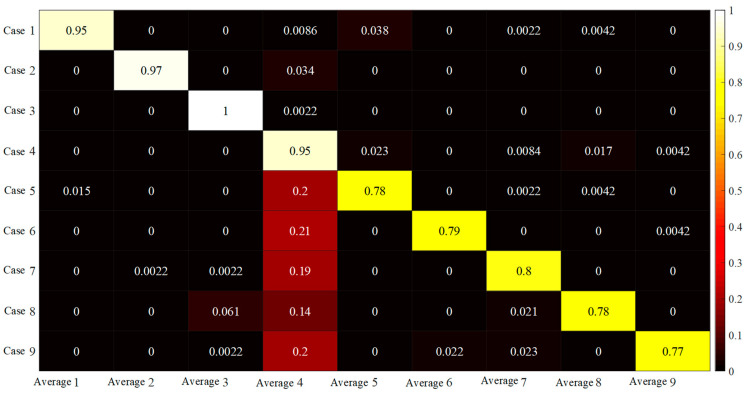
Predicted by 5 BICNN models for 9 damage cases.

**Table 1 sensors-23-00418-t001:** Nine damage cases.

Damage Cases	Specific Operation
1	No damage
2	Remove structures numbered 1–8 in the east
3	Remove structures numbered 1–4 in the east
4	Remove structures numbered 1 and 4 in the east
5	Remove structure numbered 1 in the east
6	Remove structures numbered 3 and 7 in the north
7	Remove structures numbered 1–8 in all 4 sides
8	Loosen structures numbered 9, 10, 11 and 12 in the east on the basis of case 7
9	Loosen structures numbered 11 and 12 in the east on the basis of case 7

**Table 2 sensors-23-00418-t002:** Sample size.

Sensor	Train Set	Test Set	Reserved Data
4~15	3863	1655	1380

**Table 3 sensors-23-00418-t003:** Specific parameters of BICNN.

No.	Layer Type	Kernel Size/Stride	Kernel Channel Size	Output Size (Width × Depth)	Padding
1	Conv_1	64 × 1/8 × 1	16	128 × 16	yes
2	Max pooling	2 × 1/2 × 1	16	64 × 16	yes
3	Conv_2	32 × 1/1 × 1	32	64 × 32	yes
4	Max pooling	2 × 1/2 × 1	32	32 × 32	yes
5	Inception_1		64	32 × 64	
6	Max pooling	2 × 1/2 × 1	64	16 × 64	yes
7	Inception_2		64	16 × 64	
8	Max pooling	2 × 1/2 × 1	64	8 × 64	yes
9	Inception_3		64	8 × 64	
10	Max pooling	2 × 1/2 × 1	64	4 × 64	yes
11	Inception_4		64	4 × 64	
12	Max pooling	2 × 1/2 × 1	64	2 × 64	yes
13	Inception_5		64	2 × 64	
14	Max pooling	2 × 1/2 × 1	64	1 × 64	yes
15	Dense_1	100	1	100 × 1	
16	Dense_2	50	1	50 × 1	
17	Softmax	9	1	9	

**Table 4 sensors-23-00418-t004:** Experimental results of optimizing training samples.

Sensor ID	1DCNN	WDCNN	TDCNN	ITICNN	BICNN
4	40.36%	99.46%	99.88%	99.96%	99.99%
5	55.38%	99.77%	99.73%	100%	100%
6	49.21%	100%	100%	99.77%	99.99%
7	50.11%	90.08%	90.34%	91.33%	93.98%
8	21.22%	84.00%	76.92%	85.31%	90.01%
9	42.75%	99.65%	99.96%	100%	100%
10	26.21%	84.38%	77.70%	83.46%	89.63%
11	29.15%	84.77%	78.35%	85.23%	88.74%
12	32.35%	84.07%	78.04%	84.85%	90.23%
13	35.79%	85.23%	79.05%	85.31%	90.38%
14	47.43%	99.61%	99.81%	99.73%	100%
15	59.03%	83.42%	78.16%	85.04%	87.56%
Number of excellent sensors	0	5	5	5	5

**Table 5 sensors-23-00418-t005:** Diagnosis accuracy of five neural network models in noise environment.

k	1DCNN	WDCNN	TICNN	ITICNN	BICNN
0	100%	100%	100%	100%	100%
0.1	100%	100%	100%	100%	100%
0.2	94.10%	83.31%	84.82%	99.8%	99.9%
0.3	66.63%	47.65%	54.25%	91.31%	95.8%
0.4	54.35%	32.47%	35.46%	72.93%	87.21%
0.5	50.25%	24.68%	26.37%	55.84%	77.03%
0.6	46.35%	22.38%	22.88%	43.56%	59.75%
0.7	41.86%	19.58%	22.48%	32.37%	50.66%
0.8	37.66%	19.78%	22.48%	28.97%	43.77%
0.9	34.47%	18.98%	23.58%	26.07%	39.27%
1	32.77%	18.18%	22.97%	23.28%	36.17%

**Table 6 sensors-23-00418-t006:** Integrated diagnosis accuracy of five convolutional neural networks.

k	1DCNN	WDCNN	TICNN	ITICNN	BICNN
0	38.36%	64.93%	46.08%	93.51%	97.38%
0.1	38.36%	64.69%	46.08%	93.51%	97.38%
0.2	32.46%	48.00%	30.9%	93.31%	97.28%
0.3	24.99%	42.34%	20.33%	84.82%	93.18%
0.4	17.29%	38.84%	18.46%	66.44%	84.59%
0.5	11.39%	30.63%	7.55%	49.35%	74.41%
0.6	3.22%	19.93%	3.04%	37.07%	57.13%
0.7	1.1%	15.26%	1.44%	25.88%	48.04%
0.8	0%	5.53%	0%	22.48%	41.15%
0.9	0%	0%	0%	19.58%	36.65%
1	0%	0%	0%	16.79%	33.55%

## Data Availability

Data sharing not applicable.

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
