# Peer review of "Integrated Damage Location Diagnosis of Frame Structure Based on Convolutional Neural Network with Inception Module"

_sensors, 2022, doi:10.3390/s23010418_

Round 1
Reviewer 1 Report
The research topic is interesting as it enhances the diagnosis of the damaged locations of frame structures.
Overall the research is well-conducted and written.
Please consider addressing the following comments to enhance your manuscript further:
· The Abstract is a bit long. Try to reduce it to increase its readability.
· Avoid using Pronouns. They are not common in research papers. For instance, you used "we" 7 times in the manuscript.
· Line 124 seismic or Seismic? Also, provide more information about the building to give the reader more information about its context.
· Provide the meanings of the abbreviations used in Figure 2 and equation 2 as a note under them to facilitate exploring your manuscript, and please check if this is needed in other locations.
· Improve the presentation of Figures 3 and 4.
· Check the structure of Table 3. It is hard to read. Furthermore, adjust the columns' width of Table 4 to fit the text better.
· The discussion needs to be improved, and compare your results with relative literature and cite this literature where it is needed. The current form indicates as you are only reporting your improvement approach.
Best of luck.
Author Response
Thank you very much for your comments. Attached is our detailed reply.

Reviewer 2 Report
Overall, the paper is very well presented. Adequate explanations are provided to justify the results. I have two minor concerns as follows:
(1) In your introduction of the noise environment, you use a k-value to indicate the severity of the noise. It would be useful if the authors provide examples of what a noise level of k = 1.0 represents versus k = 0.5 so that the reader can understand the relative noise levels introduced.
(2) In Figure 8, the authors conclude that the data collected by the sensor number 4 are faulty and were removed to improve the prediction levels. This process seems random. How are users supposed to know what sensors to be deleted and what should be kept. Also, what are the chances of the sensor being faulty or incorrectly mounted? It would be nice to explore the reasons behind the removal of sensor 4.
Author Response
Thank you very much for your comments. Attached is our detailed reply

Reviewer 3 Report
This paper presents a 1D deep learning model approach for solving the inverse problem of structural damage detection. The authors used the time response signal of the ASCE frame model as input data for the 1DCNN model using a sliding window to provide a large number of samples for training and testing purposes. The paper is interesting and relevant and can be accepted after taking care of the following problems as
1- What are the parameter settings of Gaussian low pass filters for dealing with noise elimination?
2- Please inform the readers about the Cij in Eq. (3).
3- What are the levels of noise added to the data? I know the model very well and the level of noise can be set up easily.
4- Please provide a full schematic diagram of the full process including the integration of noise handling, 1DCNN, the inception model, etc. This can be done with little effort and can be more vivid for reader understanding.
5- On what basis the parameters are selected in Table 3? Please discuss!
6- The abstract should be improved, rewritten, and include clear statements about objectives, methodology, and findings. It should not be served as an introduction to the problem.
7- The introduction needs to be improved. The model has been used to study the inverse problems of damage identification by various researchers, which the authors have missed.
Author Response

(The authors gave the same response as above.)

Round 2
Reviewer 1 Report
The authors have made significant improvements and addressed the provided comments.
I support accepting the manuscript in its current form.
Reviewer 3 Report
The paper can be accepted for publication.